# COVID-19 on Chest CT: Translating Known Microscopic Findings to Imaging Observations

**DOI:** 10.3390/life12060855

**Published:** 2022-06-08

**Authors:** Belinda Dsouza, Kathleen M. Capaccione, Aron Soleiman, Jay Leb, Mary Salvatore

**Affiliations:** 1Department of Radiology, Columbia University Medical Center, New York, NY 10032, USA; bmd2111@cumc.columbia.edu (B.D.); kmc9020@nyp.org (K.M.C.); jl4763@cumc.columbia.edu (J.L.); 2Department of Medicine, Montefiore Medical Center, Bronx, NY 10467, USA; asoleiman@montefiore.org

**Keywords:** COVID-19, chest CT, Epithelial, endothelial, infarct

## Abstract

Purpose: To describe the imaging findings of COVID-19 and correlate them with their known pathology observations. Methods: This is an IRB-approved retrospective study performed at Columbia University Irving Medical Center (*IRB # AAAS9652*) that included symptomatic adult patients (21 years or older) who presented to our emergency room and tested positive for COVID-19 and were either admitted or discharged with at least one chest CT from 11 March 2020 through 1 July 2020. CT scans were ordered by the physicians caring for the patients; our COVID-19 care protocols did not specify the timing for chest CT scans. A scoring system was used to document the extent of pulmonary involvement. The total CT grade was the sum of the individual lobar grades and ranged from 0 (no involvement) to 16 (maximum involvement). The distribution of lung abnormalities was described as peripheral (involving the outer one-third of the lung), central (inner two-thirds of the lung), or both. Additional CT findings, including the presence of pleural fluid, atelectasis, fibrosis, cysts, and pneumothorax, were recorded. Contrast-enhanced CT scans were evaluated for the presence of a pulmonary embolism, while non-contrast chest CT scans were evaluated for hyperdense vessels. Results: 209 patients with 232 CT scans met the inclusion criteria. The average age was 61 years (range 23–97 years), and 56% of the patients were male. The average score reflecting the extent of the disease on the CT was 10.2 (out of a potential grade of 16). Further, 73% of the patients received contrast, which allowed the identification of a pulmonary embolism in 21%. Of those without contrast, 33% had hyperdense vessels, which might suggest a chronic pulmonary embolism. Further, 47% had peripheral opacities and 9% had a Hampton’s hump, and 78% of the patients had central consolidation, while 28% had round consolidations. Atelectasis was, overall, infrequent at 5%. Fibrosis was observed in 11% of those studied, with 6% having cysts and 3% pneumothorax. Conclusions: The CT manifestations of COVID-19 can be divided into findings related to endothelial and epithelial injury, as were seen on prior post-mortem reports. Endothelial injury may benefit from treatments to stabilize the endothelium. Epithelial injury is more prone to developing pulmonary fibrotic changes.

## 1. Introduction

The Radiology Society of North America (RSNA) has provided guidelines for reporting imaging findings in patients with coronavirus disease 2019 (COVID-19). The typical appearance of COVID-19 includes bilateral, basilar-predominant, peripheral ground-glass and/or consolidation, and vascular enlargement (Figure 1). A crazy-paving pattern, organizing pneumonia, pleural effusions, lymphadenopathy, and cavities have also been described [1]. A pulmonary embolism is reported in less than half of the cases [1,2,3,4,5,6,7]. Defining the typical appearance of COVID-19 on imaging is helpful for early and accurate diagnosis, especially when diagnostic resources may be limited [8]. More importantly, translating radiology observations into their pathophysiologic correlates could optimize the timing of interventions to improve patient outcomes.

Post-mortem examinations provide valuable information regarding the pathology of COVID-19. Prior reports reveal two main microscopic patterns of lung disease identified in COVID-19 decedents. Intravascular fibrin and/or platelet-rich aggregates (IFPAs) were found in small and large pulmonary vasculature in as many as 84 to 90% of the lungs examined post-mortem [9,10]. Acute lung injury (ALI) was present in approximately three-quarters of the patients examined [9,10,11,12,13]. The acute (exudative) phase of ALI consists of hyaline membrane formation, and the organizing (proliferative) phase demonstrates fibroblast proliferation and acute fibrinous and organizing pneumonia (AFOP). The exudative phase was observed more often than the proliferative phase (75–76% versus 38–47%, respectively) at autopsy, but both were often present together [9,10,11,12,13].

The purpose of this study is to interpret the chest CT manifestations of COVID-19 in light of the known autopsy findings. We aim to correlate the aforementioned microscopic patterns with their radiologic equivalents and evaluate their evolution over time radiographically by identifying the date of symptoms for each chest CT. By doing so, we hope to non-invasively assess how the virus affects the lungs at various stages of the infection.

## 2. Materials and Methods

This retrospective study was approved by the internal review board of Columbia University Irving Medical Center (*IRB #*
*AAAS9652*). The requirement for informed consent for this retrospective review was waived.

### 2.1. Patients

Inclusion criteria were symptomatic adult patients (21 years or older) who presented to our emergency room and tested positive for COVID-19 and were either admitted or discharged with at least one chest CT from 11 March 2020 through 1 July 2020. The age and sex of each patient were documented. In addition, we documented the number of days since symptom onset at the time of the CT scan by review of the patient medical record. Zero days means that the patient had a CT scan on the day of onset of symptoms.

### 2.2. CT Protocol

CT scans were ordered by physicians caring for the patients; our COVID-19 care protocols did not specify the timing for chest CTs. Chest CT scans were performed using one of five commercial GE 64-slice CT scanners (GE Medical Systems, Milwaukee, WI, USA) at our site. CT acquisition parameters were as follows: 120 kVp, smart mA tube current modulation (range 80–600 mA), noise index 16, pitch 1.375:1, and table movement 35 mm/rotation. Two axial reconstructions were generated for each scan: one with adaptive statistical iterative reconstruction (ASIR) at 40% value and reconstructed slice thickness of 1.25 mm with standard kernel and the other with ASIR 40% at a slice thickness of 5 mm with lung kernel. Pulmonary embolism studies were performed using no more than 100 cc of intravenous non-ionic contrast agent *Iohexol 350 mg/mL* at an average rate of 4 cc per second per our protocol. Scanning was triggered when the average Hounsfield unit of the pulmonary artery reached 100.

### 2.3. Chest CT Evaluation

A scoring system was used to document the extent of pulmonary involvement. The lung was divided into 4 quadrants, with the right upper lobe and right middle lobe graded as one quadrant. The left upper lobe, right lower, lobe, and left lower lobe were each considered a separate quadrant. Each quadrant was graded from 0–4; Grade 0 indicated no involvement; Grade 1, 1–25% involvement; Grade 2, 25–50% involvement; Grade 3, 51–75% involvement; and Grade 4, 76–100% involvement. The total CT grade was the sum of the individual lobar grades and ranged from 0 (no involvement) to 16 (maximum involvement).

The distribution of lung abnormalities was described as peripheral (involving the outer one-third of the lung), central (inner two-thirds of the lung), or both. Discrete peripheral Hampton’s humps and round consolidations were noted. Additional CT findings, including the presence of pleural fluid, atelectasis, fibrosis, cysts, and pneumothorax, were recorded. Contrast-enhanced CT scans were evaluated for the presence of pulmonary embolism, while non-contrast chest CT scans were evaluated for hyperdense vessels.

## 3. Results

### 3.1. Patient Characteristics and General Results

In total, 209 patients with 232 CT scans met the inclusion criteria; 23 patients had two CT scans. The average age was 61 years (range 23–97 years), and 56% of the patients were male. The average score reflecting the extent of the disease on the CT was 10.2 (out of a potential grade of 16). Further, 73% of the patients received contrast, which allowed the identification of a pulmonary embolism in 21%. Of the 27% without contrast, 33% had hyperdense vessels, which might suggest a chronic pulmonary embolism, 47% had peripheral opacities, and 9% had a Hampton’s hump. Additionally, 78% of the patients had central consolidation, and 28% had round consolidations. Atelectasis was, overall, infrequent, occurring in 5% of the patients. Fibrosis was observed in 11% of those studied, with 6% having cysts and 3% pneumothorax (Table 1).

### 3.2. Chest CT Scans Performed in the Acute Phase (0–4 Days)

The average age was 60 years, and 58% were male. Pulmonary embolism, hyperdense vessels, and Hampton’s humps were present in 16%, 25%, and 23%, respectively. Peripheral opacities were very common, affecting 55% of the patients. Central consolidations and round consolidations were also frequent at 55% and 39%, respectively. Fibrotic changes were present in only one patient and were pre-existing (Table 1).

### 3.3. Chest CT Scans Were Performed in the Subacute Phase (5–12 Days)

The average age was 62 years, and 48% were males. Pulmonary embolism and hyperdense vessels occurred in 14% and 38%, respectively. Peripheral opacities were very common, affecting 60% of the patients. Central and round consolidations were also frequent at 78% and 35%, respectively. Fibrotic changes were present in 5% of the patients and likely represented pre-existing disease (Table 1).

### 3.4. Chest CT Scans Were Performed in the Persistent Phase (13–28 Days)

The average age was 62 years, and 66% were males. Pulmonary embolism and hyperdense vessels remained common at 33% and 40%, respectively. Peripheral opacities were very common, affecting 55% of the patients. Central and round consolidations were also frequent at 85% and 26%, respectively. The greatest average extent of disease was found at this stage. Fibrotic changes were more common, affecting 10% of the patients (Table 1).

### 3.5. Chest CT Scans Performed in the Chronic Phase (>28 Days)

The average age was 61 years, and 56% were males. Pulmonary embolism, hyperdense vessels, and Hampton’s humps were observed in 11%, 32%, and 8%, respectively. Peripheral opacities were less common, affecting 27% of the patients. Central and round consolidations persisted at 80% and 17%, respectively. Fibrotic changes were present in 22% of the patients (Table 1).

## 4. Discussion

The published results of post-mortem examinations identified two main microscopic patterns of lung disease: IFPAs were identified in as many as 90% of the patients and ALI in 75% of the lungs. These two patterns of disease, representing pulmonary endothelial and epithelial injury, respectively, can be diagnosed radiographically.

The CT manifestations of a pro-coagulant microenvironment caused by an endothelial injury are typically pulmonary emboli, as is seen in patients with cancer and prolonged immobility [14,15]. However, the microthrombi of COVID-19 are too small to see in many patients until the disease advances. Radiologists are more likely to identify early Hampton’s humps, a secondary finding related to a vascular insult in patients with extensive microvascular disease who may not be able to rely on the dual bronchial circulation because it too may be compromised [16].

ALI is secondary to alveolar epithelial damage caused by the viral infection. The angiotensin-converting-enzyme type 2 receptor is located on type 2 alveolar epithelial cells, making them vulnerable to viral infection with COVID-19 [17]. Epithelial cell infection with the virus leads to a cascade of events that, in some patients, culminates in ARDS [18], with its associated desquamation of pneumocytes with hyaline membrane formation and fluid accumulation within the alveoli. The extensive alveolar fluid presents initially as ground-glass opacities and progresses to consolidation as the fluid increases [19]. Clearing of consolidations with residual ground-glass opacities has been identified. Our results also suggest that pulmonary fibrosis secondary to COVID-19 infection is related to the epithelial injury given its distribution along airways and association with surfactant deficiency, as has been described in other types of fibrosis [20].

### 4.1. Endothelial Dysfunction

Endothelial cell dysfunction with COVID-19 has been described [21,22,23]. Electron microscopic and histologic evidence of the direct viral infection of endothelial ACE2 receptors in multiple organs has been reported [21,22]. Endothelial inflammation, or endotheliitis, causes both vasoconstriction and a pro-coagulant environment, predisposing to thrombus and potential organ ischemia [23]. IFPA is a histologic finding consisting of either microthrombi or microemboli and was reported in the vast majority of COVID-19 decedents, as mentioned previously. Alveolar capillary microthrombi were nine times more prevalent in COVID-19 than in influenza in post-mortem studies [8,22]. The microthrombi would be too small and peripheral to yield a positive result for a pulmonary embolism in many patients. Pulmonary embolism identification with CT has an 83% sensitivity [24], with smaller peripheral pulmonary emboli more likely to be missed.

Hampton’s hump is a radiographic sign of pulmonary infarction and is rare due to the dual pulmonary and bronchial arterial supply of the lungs. Patients with COVID-19 may be predisposed to infarction due to the concurrent viral infection of bronchial arteries [16]. Hampton’s humps were identified in 9% of our patients. This sign was most prevalent in the acute phase, which may be due to the initial involvement of pulmonary vasculature with COVID-19, which may precede lung injury [25].

A large meta-analysis showed a pooled incidence rate of PE of 16.5%, similar to our study [26]. In our study, the prevalence of a pulmonary embolism was greatest between 13 and 28 days (Figure 2). Few pulmonary emboli were diagnosed in early or late disease. This would be expected because of the time it takes to propagate a microthrombotic event into a large enough size that could be detected on a chest CT. In an autopsy series performed at our institution, a pulmonary thrombosis in large vessels was more common in patients with acute lung injury than in those without (55% versus 27%) [9]. Pulmonary emboli decrease in the chronic phase of the disease, as would be expected.

Hyperdense vasculature on a non-contrast CT scan is a radiographic sign highly diagnostic of a long-standing pulmonary embolism (Figure 3) [27,28], and 33% of the patients with a non-contrast chest CT exhibited hyperdense vessels. Similar to pulmonary emboli, this sign was most prevalent in the persistent phase of the disease. As the thrombus ages, its water content decreases, concentrating the hemoglobin and raising the CT attenuation value; this process usually requires at least 80 h to occur, explaining the low prevalence of this radiographic finding during the acute phase [29,30,31]. Hyperdense vessels persist in the chronic phase of the disease as the calcifications would not be expected to resolve.

### 4.2. Epithelial Dysfunction

Epithelial cell dysfunction in COVID-19 includes alveolar flooding due to leaky capillaries and inflammatory cell infiltrates, with the potential for progression to acute respiratory distress syndrome (ARDS) with diffuse alveolar damage and hyaline membrane formation [32]. Central consolidations representing DAD increased over time and were most prevalent in the persistent phase; they were also present in 78% of all the CT scans. In contrast, peripheral opacities diminished over time, supporting earlier endothelial injury. Infection and inflammation can spread from one alveolus to the other through the pores of Kohn, thus moving centrally over time [33].

Round consolidation, likely representing organizing pneumonia, was a feature of early disease and decreased during the persistent and chronic phase of the disease [34,35]. Atelectasis, secondary to a lack of surfactant, increased over time. The type 2 alveolar epithelial cells contain receptors for the COVID-19 virus and, when infected, can be destroyed, thus diminishing the surfactant [36] and ultimately causing basilar-predominant atelectasis, with patient improvement in the prone position [37]. Figure 4 demonstrates the evolution of lung findings over time, including evidence of epithelial dysfunction.

### 4.3. Fibrosis

Pulmonary fibrosis occurs not infrequently following severe COVID-19 infection, especially if mechanically ventilated [38]. In a study of patients hospitalized for COVID-19, chest CT scans 4 months later demonstrated nonfibrotic ground-glass opacities, fibrotic reticulations, and traction bronchiectasis [38]. We defined pulmonary fibrosis as reticulations, architectural distortion, and traction bronchiolectasis for the purpose of this study. No patient had honeycombing, which may take a longer time to develop. Pulmonary fibrosis was far more prevalent in the persistent and chronic phases of COVID-19 and associated with cyst formation and pneumothoraxes. The paucity of fibrosis earlier on is expected as extracellular matrix deposition requires time. In our patients, fibrosis was associated with older age, male sex, and a higher average extent of disease. The endothelial disease was not as frequently identified in the patients with fibrosis compared to those without. Central consolidations were, however, associated with fibrosis. The lack of alveolar epithelial cells with resultant atelectasis predisposes a patient to collapse induration with collagen deposition and irreversible fibrosis [37]. Therefore, the prevention of atelectasis may have important benefits in the prevention of fibrosis.

Efferocytosis is the removal of apoptotic cells by macrophages. Impaired efferocytosis is associated with many pulmonary diseases, including cystic fibrosis, chronic obstructive pulmonary disease, bronchiectasis, and asthma [39]. A recent study by dos-Santos et al. showed that engulfment of COVID-19-infected apoptotic cells exacerbated inflammatory cytokine production and inhibited efferocytotic receptors, with decreased macrophage phagocytosis [40]. Impaired efferocytosis could contribute to increased fibrosis in COVID-19. Efferocytosis in the older male patients, with comorbidities, may be relatively dysfunctional at the baseline, resulting in more fibrosis, as seen in our study.

The cause of cyst formation in COVID-19 is unknown [41]; however, peripherally located cysts could represent an infarcted lung, and central cysts could represent pneumatoceles [42]. It is not surprising that these would increase in late disease and be associated with an increase in pneumothoraxes. In addition, stiff lung parenchyma from persistent infection and mechanical ventilation can predispose patients to pneumothoraxes [43,44] (Figure 5). Table 2 describes the characteristics of patients who developed fibrosis due to COVID-19 infection.

## 5. Conclusions

The results of our study support the importance of both an endothelial and epithelial injury in many patients with severe COVID-19 infection and that a CT scan can facilitate the differentiation of the primary site of involvement and its evolution over time, which has implications for the management of patients.

## Figures and Tables

**Figure 1 life-12-00855-f001:**
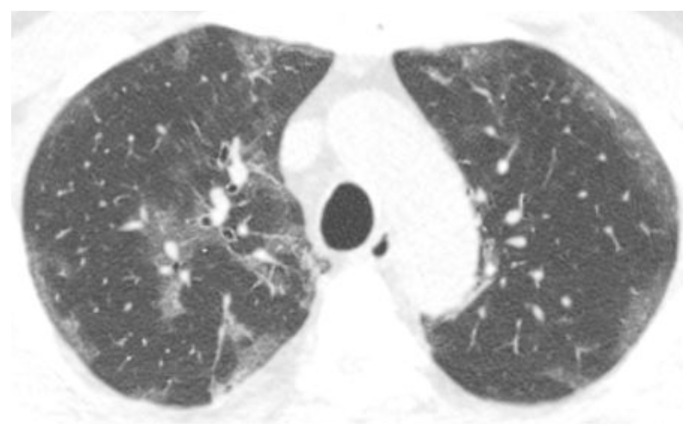
Axial chest CT image demonstrates peripheral ground-glass opacities in a patient with the early phase of COVID-19 infection.

**Figure 2 life-12-00855-f002:**
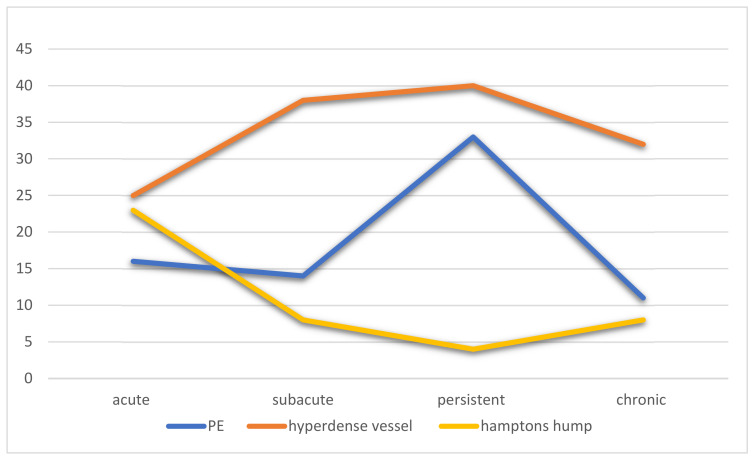
Percentage of patients over time with findings related to endothelial injury, including pulmonary embolism, hyperdense vessel, and Hampton’s hump.

**Figure 3 life-12-00855-f003:**
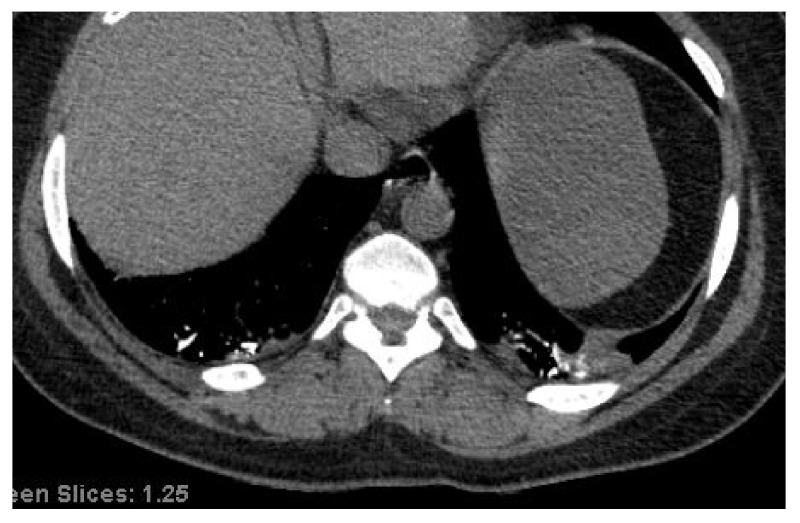
Non-contrast chest CT on mediastinal window settings in a patient with chronic COVID-19 demonstrates hyperdense vessels compatible with clotted blood from thrombotic disease.

**Figure 4 life-12-00855-f004:**
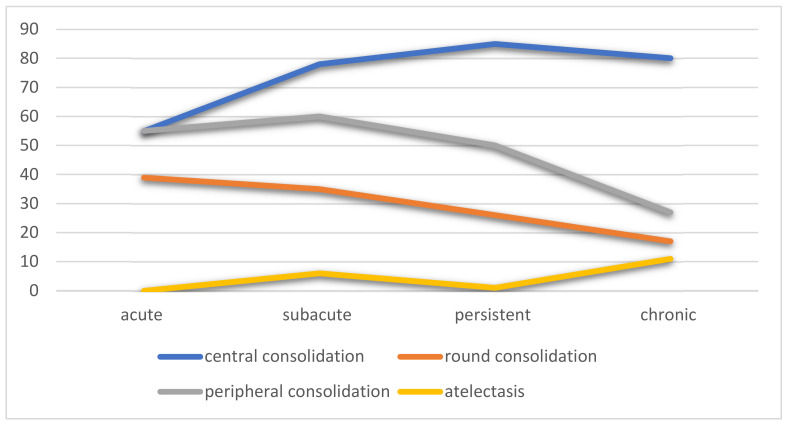
Percentage of patients over time with findings related to epithelial injury, including central consolidation, round consolidation, peripheral consolidation, and atelectasis.

**Figure 5 life-12-00855-f005:**
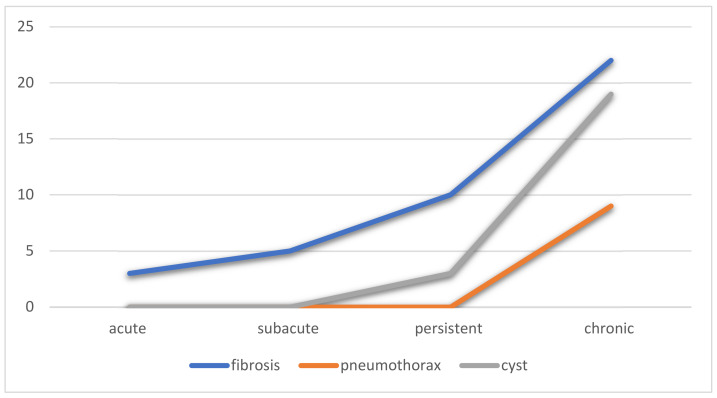
Percentage of patients over time with findings related to fibrosis, including fibrosis, pneumothorax, and cysts.

**Table 1 life-12-00855-t001:** Chest CT findings in patients with COVID-19.

	Acute (0–4 Days)	Subacute (5–12 Days)	Persistent (13–28 Days)	Chronic (>28 Days)	Total (0–249 Days)
**Number of CT scans**	31	63	74	64	232
** *Demographics* **					
Average age (range)	60 (23–97)	62 (27–88)	62 (28–87)	61 (25–85)	61 (23–97)
% male	18/31 (58%)	30/63 (48%)	49/74 (66%)	32/64 (50%)	129/232 (56%)
The extent of disease (0–16)	7.7	9.8	10.8	11.0	10.2
** *Endothelial disease* **					
Contrast	19/31 (61%)	50/63 (79%)	64/74 (86%)	36/64 (56%)	169/232 (73%)
Pulmonary embolism	3/19 (16%)	7/50 (14%)	21/64 (33%)	4/36 (11%)	35/169 (21%)
Hyperdense vessel	3/12 (25%)	5/13 (38%)	4/10 (40%)	9/28 (32%)	21/63 (33%)
Hampton’s hump	7/31 (23%)	5/63 (8%)	3/74 (4%)	5/64 (8%)	20/232 (9%)
Effusion	2/31 (6%)	7/63 (11%)	11/74 (15%)	15/64 (23%)	35/232 (15%)
** *Epithelial disease* **					
Central consolidation	17/31 (55%)	49/63 (78%)	63/74 (85%)	51/64 (80%)	180/232 (78%)
Round consolidation	12/31 (39%)	22/63 (35%)	19/74 (26%)	11/64 (17%)	64/232 (28%)
Peripheral opacity	17/31 (55%)	38/63 (60%)	37/74 (50%)	17/64 (27%)	109/232 (47%)
Atelectasis	0/31 (0%)	4/63 (6%)	1/74 (1%)	7/64 (11%)	12/232 (5%)
** *Scarring* **					
Fibrosis	1/31 (3%)	3/63 (5%)	7/74 (10%)	14/64 (22%)	25/232 (11%)
Pneumothorax	0/31 (0%)	0/63 (0%)	0/74 (0%)	6/64 (9%)	6/232 (3%)
Cysts	0/31 (0%)	0/63 (0%)	2/74 (3%)	11/64 (17%)	13/232 (6%)

**Table 2 life-12-00855-t002:** CT findings in patients with fibrosis.

Characteristics	Fibrosis (n = 25)	No Fibrosis (n = 207)	*p*-Value
Average age	64	61	0.42
% male	17/25 (68%)	112/207 (54%)	0.19
The average extent of disease	11.8	10.0	*0.04*
Pulmonary embolism	1/15 (7%)	34/154 (22%)	0.19
Hyperdense vessel	6/25 (24%)	21/55 (38%)	0.21
Peripheral opacity	5/25 (20%)	104/207 (50%)	*0.004*
Hampton’s hump	1/25 (4%)	19/207 (9%)	0.39
Central consolidation	22/25 (88%)	158/207 (76%)	0.19
Round consolidation	5/25 (20%)	59/207 (29%)	0.19
Effusion	4/25 (16%)	31/207 (15%)	0.89
Pneumothorax	4/25 (16%)	2/207 (1%)	*<0.00001*
Cysts	4/25 (16%)	9/207 (4%)	*0.02*

## Data Availability

De-identified data can be made available.

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
