# Peer review of "COVID-19 on Chest CT: Translating Known Microscopic Findings to Imaging Observations"

_life, 2022, doi:10.3390/life12060855_

Round 1

Reviewer 1 Report

  1. In abstract, please add study design, study place, study duration etc because it is a research abstract and it must fulfill the criteria to write an abstract.

2.Please clearly describe the four lobes of lung for clear understanding of the readers.

3.Please clearly describe how to interpret the epithelial and endothelial injury in CT scan of lung including the reference for diagnosing it.

4.All patients were doing post mortem examination? It is not clear because the authors would like to say epithelial and endothelial injury and correlate with CT scan findings. How do you correlate without histological examination?

  1. If the authors did post mortem examination, how about the consent taking from the guardians?
  2. This study did not get ethical approval from any institutional review board or ethics review committees? The reviewer dd not see any information about ethical certificate. Please add at the revised one or difficult to publish without ethical approval because this study involve human subjects although it is a retrospective study.
  3. Informed consent is an important for conducting study involving human subject. It is a retrospective study and if the researchers dis not take consent please describe the reasons why the authors did not take consent in this study. Please add at the revised manuscript.

Author Response

Thank you sincerely for your review of our manuscript and your helpful insights to improve the quality of the manuscript. I have attached a response for each of your insights below.

In the abstract, please add study design, study place, study duration, etc. because it is a research abstract and it must fulfill the criteria to write an abstract.

This is an IRB-approved, retrospective study performed at Columbia University Irving Medical Center. Symptomatic adult patients (21 years or older) who presented to our emergency room and tested positive for COVID-19 and either admitted or discharged with at least one chest CT from March 11, 2020, through July 1, 2020.

Please clearly describe the four lobes of the lung for a clear understanding for the readers

The lung was divided into 4 quadrants with the right upper lobe and right middle lobe graded as one quadrant. The left upper lobe, right lower, lobe, and left lower lobe were each considered a separate quadrant

Please clearly describe how to interpret the epithelial and endothelial injury in a CT scan of the lung including the reference for diagnosing it.

The CT manifestations of a pro-coagulant microenvironment caused by an endothelial injury are typically pulmonary emboli as is seen in patients with cancer and prolonged immobility. (Doherty S. Pulmonary embolism an update. Aust Fam Physician. 2017 Nov;46(11):816-820. PMID: 29101916.). However, the microthrombi of COVID-19 would be too small to see in many patients and would more likely cause Hampton’s humps, a secondary finding related to a vascular insult (Salvatore MM. Pulmonary infarcts in COVID-19. Clin Imaging. 2021 Dec; 80:158-159).

ALI is secondary to alveolar epithelial damage caused by the COVID-19 virus. The angiotensin-converting enzyme type 2 receptor located on the Type 2 alveolar epithelial cells making them vulnerable to viral infection with COVID-19 (Mulay A, Konda B, Garcia G Jr, Yao C, Beil S, Villalba JM, Koziol C, Sen C, Purkayastha A, Kolls JK, Pociask DA, Pessina P, de Aja JS, Garcia-de-Alba C, Kim CF, Gomperts B, Arumugaswami V, Stripp BR. SARS-CoV-2 infection of primary human lung epithelium for COVID-19 modeling and drug discovery. Cell Rep. 2021 May 4;35(5):109055.). Epithelial cell infection with the virus leads to a cascade of events that in some patients culminates in ARDS, (Matthay MA, Zimmerman GA. Am J Respir Cell Mol Biol. 2005;33:319-27) with its associated desquamation of pneumocytes with hyaline membrane formation and fluid accumulation within the alveoli. The extensive alveolar fluid presents initially as ground-glass opacities and progresses to consolidation as the fluid increases (Battista G, Sassi C, Zompatori M, Palmarini D, Canini R. Ground-glass opacity: interpretation of high-resolution CT findings. Radiol Med. 2003 Nov-Dec;106(5-6):425-42; quiz 443-4.) Clearing of consolidations with residual ground-glass opacities have been identified. Our results also suggest that pulmonary fibrosis secondary to COVID-19 infection is related to the epithelial injury given its distribution along airways and association with surfactant deficiency as has been described in other types of fibrosis 32.

All patients were doing post-mortem examination? It is not clear because the authors would like to say epithelial and endothelial injury and correlate with CT scan findings. How do you correlate without histological examination?

We have attempted to translate the two main findings on pathology which were microembolic disease and acute lung injury into how they would appear on CT. The majority of patients who had autopsies were too sick to have CT scans.

If the authors did post mortem examination, how about the consent taken from the guardians?

We did not use post-mortem exams for this project.

This study did not get ethical approval from any institutional review board or ethics review committees? The reviewer did not see any information about the ethical certificate. Please add the revised one or difficult to publish without ethical approval because this study involves human subjects although it is a retrospective study.

We obtained IRB approval (AAAS9652) for this retrospective study from Columbia University Medical Center’s IRB. Consent was waived for this retrospective study.

Informed consent is important for conducting a study involving human subject. It is a retrospective study and if the researchers did not take consent please describe the reasons why the authors did not take consent in this study. Please add to the revised manuscript.

We obtained IRB approval (AAAS9652) for this retrospective study from Columbia University Medical Center’s IRB. Consent was waived for this retrospective study.

Reviewer 2 Report

My main suggestion is to provide, if possible, direct correlations between post-mortem microscopic findings and CT signs for each patient. This association as detected in the study population was just mentioned in passing at the beginning of the Discusson section, yet it should be the core topic of the manuscript, in line with the title.

Minor comments:

a) Did all patients undergo a post-mortem examination? If not, which proportion of them? Please also report the main demographic and clinical features of this subset population.

b) Please describe in more detail the contrast medium injection protocol used in those patients who underwent contrast-enhanced CT (including median and range of contrast volume, flow rate, etc.). Please also replace the commercial term "Omnipaque 350" with "iohexol 350mgI/mL".

Author Response

Thank you sincerely for your review of our manuscript and your helpful insights to improve the quality of the manuscript. I have attached a response for each of your insights below.

My main suggestion is to provide, if possible, direct correlations between post-mortem microscopic findings and CT signs for each patient. This association as detected in the study population was just mentioned in passing at the beginning of the Discussion section, yet it should be the core topic of the manuscript, in line with the title.

Published results of post-mortem examinations identified two main microscopic patterns of lung disease: IFPAs were identified in as many as 90% of patients and ALI in 75% of lungs. These two patterns of disease, representing pulmonary endothelial and epithelial injury respectively can be diagnosed radiographically.

 The CT manifestations of a pro-coagulant microenvironment caused by an endothelial injury are typically pulmonary emboli as is seen in patients with cancer and prolonged immobility. (Doherty S. Pulmonary embolism an update. Aust Fam Physician. 2017 Nov;46(11):816-820. PMID: 29101916.). (Iba T, Connors JM, Levy JH. The coagulopathy, endotheliopathy, and vasculitis of COVID-19. Inflamm Res. 2020 Dec;69(12):1181-1189.) However, the microthrombi of COVID-19 are too small to see in many patients until the disease advances. Radiologists are more likely to identify Hampton’s humps, a secondary finding related to a vascular insult in patients with extensive microvascular disease who may not be able to rely on the dual bronchial circulation because it too may be compromised (Salvatore MM. Pulmonary infarcts in COVID-19. Clin Imaging. 2021 Dec; 80:158-159).

ALI is secondary to alveolar epithelial damage caused by the virus. The angiotensin-converting-enzyme type 2 receptor located on the Type 2 alveolar epithelial cells making them vulnerable to viral infection with COVID-19 (Mulay A, Konda B, Garcia G Jr, Yao C, Beil S, Villalba JM, Koziol C, Sen C, Purkayastha A, Kolls JK, Pociask DA, Pessina P, de Aja JS, Garcia-de-Alba C, Kim CF, Gomperts B, Arumugaswami V, Stripp BR. SARS-CoV-2 infection of primary human lung epithelium for COVID-19 modeling and drug discovery. Cell Rep. 2021 May 4;35(5):109055.). Epithelial cell infection with the virus leads to a cascade of events that in some patients culminates in ARDS (Matthay MA, Zimmerman GA. Am J Respir Cell Mol Biol. 2005;33:319-27) with its associated desquamation of pneumocytes with hyaline membrane formation and fluid accumulation within the alveoli. The extensive alveolar fluid presents initially as ground-glass opacities and progresses to consolidation as the fluid increases (Battista G, Sassi C, Zompatori M, Palmarini D, Canini R. Ground-glass opacity: interpretation of high-resolution CT findings. Radiol Med. 2003 Nov-Dec;106(5-6):425-42; quiz 443-4.) Clearing of consolidations with residual ground-glass opacities have been identified. Our results also suggest that pulmonary fibrosis secondary to COVID-19 infection was related to the epithelial injury given its distribution along airways and association with surfactant deficiency as has been described in other types of fibrosis 32.

Minor comments:

  1. a) Did all patients undergo a post-mortem examination? If not, which proportion of them? Please also report the main demographic and clinical features of this subset population.

The study only looks at the radiology of patients with COVID-19 and attempts to translate to the known post-mortem findings we identified in a prior paper that did not have CT scans.

  1. b) Please describe in more detail the contrast medium injection protocol used in those patients who underwent contrast-enhanced CT (including median and range of contrast volume, flow rate, etc.). Please also replace the commercial term "Omnipaque 350" with "iohexol 350mgI/mL".

Pulmonary embolism studies were performed using no more than 100cc of intravenous non-ionic contrast agent Iohexol 350mg/ml at an average rate of 4cc per second per our protocol. Scanning was triggered when the average Hounsfield Unit of the pulmonary artery reached 100. 

Round 2

Reviewer 1 Report

I have no more comments.

Reviewer 2 Report

Ok. No further comments.